# The relationship between circulating lipids and breast cancer risk: A Mendelian randomization study

Kelsey E. Johnson[ID][1☯], Katherine M. Siewert[ID][2☯], Derek Klarin[ID][3,4,5], Scott M. Damrauer[ID][6,7], the VA Million Veteran Program[¶], Kyong-Mi Chang[ID][6,8], Philip S. Tsao[ID][9,10], Themistocles L. Assimes[ID][9,10], Kara N. Maxwell[ID][8,11], Benjamin F. Voight[ID][6,11,12,13]*

**1** Cell and Molecular Biology Graduate Group, University of Pennsylvania Perelman School of Medicine, Philadelphia, Pennsylvania, United States of America, **2** Genomics and Computational Biology Graduate Group, University of Pennsylvania Perelman School of Medicine, Philadelphia, Pennsylvania, United States of America, **3** Boston VA Healthcare System, Boston, Massachusetts, United States of America, **4** Center for Genomic Medicine, Massachusetts General Hospital, Harvard Medical School, Boston, Massachusetts, United States of America, **5** Program in Medical and Population Genetics, Broad Institute of MIT and Harvard, Cambridge, Massachusetts, United States of America, **6** Corporal Michael Crescenz VA Medical Center, Philadelphia, Pennsylvania, United States of America, **7** Department of Surgery, Perelman School of Medicine, University of Pennsylvania, Philadelphia, Pennsylvania, United States of America, **8** Department of Medicine, Perelman School of Medicine, University of Pennsylvania, Philadelphia, Pennsylvania, United States of America, **9** VA Palo Alto Health Care System, Palo Alto, California, United States of America, **10** Department of Medicine, Stanford University School of Medicine, Stanford, California, United States of America, **11** Department of Genetics, Perelman School of Medicine, University of Pennsylvania, Philadelphia, Pennsylvania, United States of America, **12** Department of Systems Pharmacology and Translational Therapeutics, Perelman School of Medicine, University of Pennsylvania, Philadelphia, Pennsylvania, United States of America, **13** Institute for Translational Medicine and Therapeutics, Perelman School of Medicine, University of Pennsylvania, Philadelphia, Pennsylvania, United States of America

☯ These authors contributed equally to this work.
¶ Membership of the VA Million Veteran Program is provided in S1 Text.
* bvoight@pennmedicine.upenn.edu

**Data Availability Statement:** The summary statistics for the MR instrumental variables are available in S1, S2 and S3 Tables. Genome-wide

## Abstract

### Background

A number of epidemiological and genetic studies have attempted to determine whether levels of circulating lipids are associated with risks of various cancers, including breast cancer (BC). However, it remains unclear whether a causal relationship exists between lipids and BC. If alteration of lipid levels also reduced risk of BC, this could present a target for disease prevention. This study aimed to assess a potential causal relationship between genetic variants associated with plasma lipid traits (high-density lipoprotein, HDL; low-density lipoprotein, LDL; triglycerides, TGs) with risk for BC using Mendelian randomization (MR).

### Methods and findings

Data from genome-wide association studies in up to 215,551 participants from the Million Veteran Program (MVP) were used to construct genetic instruments for plasma lipid traits.

summary statistics are available from the Global Lipids Genetics Consortium (GLGC) at http://csg. sph.umich.edu/abecasis/public/lipids2013/ and the Breast Cancer Association Consortium (BCAC) at http://bcac.ccge.medschl.cam.ac.uk/bcacdata/ oncoarray/oncoarray-and-combined-summary-result/gwas-summary-results-breast-cancer-risk-2017/. The Million Veterans Program (MVP) lipid GWAS results are available in dbGAP. The dbGAP accession number for MVP overall is phs001672. v3.p1. The accession numbers for the European-specific MVP data are TC: pha004834.1, LDL: pha004831.1, HDL: pha004828.1, and TG: pha004837.1. BMI summary statistics from Yengo et al. are available at https://portals.broadinstitute. org/collaboration/giant/index.php/GIANT_ consortium_data_files#2018_GIANT_and_UK_ BioBank_Meta-analysis. Age of menarche summary statistics from Day et al are available at https://www.reprogen.org/data_download.html. The UK10K data utilized in the study cannot be shared publicly (per data use access agreement) but are available by Institutional Data Access request for researchers who meet the criteria for access at https://www.sanger.ac.uk/legal/DAA/ MasterController.

**Funding:** This work was supported by the US National Institutes of Health (R01 DK101478 and HG010067 for BFV, T32 GM008216 for KEJ, T32 HG000046 for KMS) and a Linda Pechenik Montague Investigator award (to BFV). This research is based on data from the Million Veteran Program, Office of Research and Development, Veterans Health Administration and was supported by award #MVP000. This research was also supported by two additional Department of Veterans Affairs awards (I01-BX003362 [PST/KC], IK2-CX001780 [Damrauer]). This publication does not represent the views of the Department of Veterans Affairs or the United States Government. This study makes use of data generated by the UK10K Consortium, derived from samples from the Avon Longitudinal Study of Parents and Children (ALSPAC) and the Department of Twin Research and Genetic Epidemiology (DTR), the TWINSUK Cohort. A full list of the investigators who contributed to the generation of the data is available from www.UK10K.org. Funding for UK10K was provided by the Wellcome Trust under award WT091310. The funders had no role in study design, data collection and analysis, decision to publish, or preparation of the manuscript.

**Competing interests:** I have read the journal's policy and the authors of this manuscript have the following competing interests: SMD declares research support to institution from Renalytix AI

The effect of these instruments on BC risk was evaluated using genetic data from the BCAC (Breast Cancer Association Consortium) based on 122,977 BC cases and 105,974 controls. Using MR, we observed that a 1-standard–deviation genetically determined increase in HDL levels is associated with an increased risk for all BCs (HDL: OR [odds ratio] = 1.08, 95% confidence interval [CI] = 1.04–1.13, P < 0.001). Multivariable MR analysis, which adjusted for the effects of LDL, TGs, body mass index (BMI), and age at menarche, corroborated this observation for HDL (OR = 1.06, 95% CI = 1.03–1.10, P = 4.9 × 10$^{-4}$) and also found a relationship between LDL and BC risk (OR = 1.03, 95% CI = 1.01–1.07, P = 0.02). We did not observe a difference in these relationships when stratified by breast tumor estrogen receptor (ER) status. We repeated this analysis using genetic variants independent of the leading association at core HDL pathway genes and found that these variants were also associated with risk for BCs (OR = 1.11, 95% CI = 1.06–1.16, P = 1.5 × 10$^{-6}$), including locus-specific associations at *ABCA1* (ATP Binding Cassette Subfamily A Member 1), *APOE-APOC1-APOC4-APOC2* (Apolipoproteins E, C1, C4, and C2), and *CETP* (Cholesteryl Ester Transfer Protein). In addition, we found evidence that genetic variation at the ABO locus is associated with both lipid levels and BC. Through multiple statistical approaches, we minimized and tested for the confounding effects of pleiotropy and population stratification on our analysis; however, the possible existence of residual pleiotropy and stratification remains a limitation of this study.

## Conclusions

We observed that genetically elevated plasma HDL and LDL levels appear to be associated with increased BC risk. Future studies are required to understand the mechanism underlying this putative causal relationship, with the goal of developing potential therapeutic strategies aimed at altering the cholesterol-mediated effect on BC risk.

## Author summary

### Why was this study done?

- An individual's lipid levels may affect their risk of breast cancer. However, previous studies disagree on whether a causal effect exists.

- Mendelian randomization methods allow researchers to test whether genetically influenced lipid levels are associated with risk of breast cancer.

### What did the researchers do and find?

- We tested whether genetic variants that are associated with changes in lipid levels also have consistent associations with breast cancer.

- We found that both high and low-density lipoprotein cholesterol (HDL and LDL) are associated with an increased risk of breast cancer.

and a patent application filed by VA on drug repurposing for lipid reduction.

**Abbreviations:** *ABCA1*, ATP Binding Cassette Subfamily A Member 1; *APOC*, Apolipoprotein C1; *APOE*, Apolipoprotein E; BC, breast cancer; BCAC, Breast Cancer Association Consortium; BMI, body mass index; *CETP*, Cholesteryl Ester Transfer Protein; CI, confidence interval; ER, estrogen receptor; GLGC, Global Lipids Genetics Consortium; GWAS, genome-wide association study; HDL, high-density lipoprotein; *HMGCR*, 3-Hydroxy-3-Methylglutaryl-CoA Reductase; *LCAT*, Lecithin-Cholesterol Acyltransferase; LD, linkage disequilibrium; LDL, low-density lipoprotein; *LDLR*, LDL Receptor; *LIPC*, Lipase C, Hepatic Type; LIPG, Lipase G, Endothelial Type; *LPA*, Lipoprotein(A); MR, Mendelian randomization; MVP, Million Veteran Program; *MYLIP*, Myosin Regulatory Light Chain Interacting Protein; *NPC1L1*, NPC1-Like Intracellular Cholesterol Transporter 1; OR, odds ratio; *PCSK9*, Proprotein Convertase Subtilisin/Kexin Type 9; *PLTP*, Phospholipid Transfer Protein; *SCARB1*, Scavenger Receptor Class B Member 1; STROBE, Strengthening the Reporting of Observational Studies in Epidemiology; TC, total cholesterol; TG, triglyceride.

## What do these findings mean?

- The techniques used in this study cannot rule out that our findings are due to the lipid-associated genetic variants being associated with breast cancer risk through mechanisms other than cholesterol level.

- Further research will be needed to investigate the possibility that manipulation of LDL or HDL levels can influence risk of breast cancer.

## Introduction

Breast cancer (BC) is the second leading cause of death for women, motivating the need for a better understanding of its etiology and more effective treatments [1]. Cholesterol is a known risk factor for multiple diseases that have reported associations with BC, including obesity, heart disease, and diabetes [2]. However, it is unknown whether cholesterol plays a causal role in BC susceptibility.

The body of epidemiological and clinical trial studies to date has yet to determine whether there is a causal relationship between cholesterol and BC. Observational epidemiological studies have reported positive, negative, or no relationship between lipid levels and BC risk [3–6]; however, these studies can suffer from confounding. A comprehensive meta-analysis found evidence that statin use may reduce BC risk [7], and cholesterol-lowering medications have been associated with improved outcomes in BC patients on hormonal therapy, suggesting an interaction of circulating cholesterol levels with estrogen-sensitive breast tissues [8]. These mixed findings motivate the need for a high-powered causal inference analysis of lipids on BC.

To try to resolve these discrepancies, recent studies have applied the framework of Mendelian randomization (MR) to determine whether genetically elevated lipid levels associate with BC risk. In a small sample of 1,187 BC cases, Orho-Melander and colleagues used multivariable MR to find suggestive evidence of a relationship between both triglycerides and HDL (high-density lipoprotein) cholesterol and BC, but no association between LDL (low-density lipoprotein) cholesterol and cancer [9]. In a second study, Nowak and colleagues [10] performed an MR analysis with genetic association data from large genome-wide association studies (GWASs) for lipids and BC [11,12]. They reported nominal positive associations between LDL-cholesterol levels and all BCs and between HDL-cholesterol levels and ER (estrogen receptor)-positive BCs. While compelling, this study also had limitations. First, they used relatively few variants in their genetic instrument because of the removal of pleiotropic variants in order to address confounding due to pleiotropy, resulting in a conservative analysis. Second, they analyzed each lipid trait separately rather than take advantage of multivariable methods to consider lipid traits together along with additional, potentially confounding causal risk factors. Third, the authors did not quantitatively assess heterogeneity to determine whether the observed lipid associations were statistically different across BC subtypes. Another recent study by Qi and Chatterjee applied a newly developed MR method and reported an association between HDL-cholesterol and BC that they defined as borderline statistically significant [13]. Like Nowak and colleagues, this paper also does not explicitly include correlated risk factors in their analysis and did not stratify BC by ER status.

These studies motivate an MR study that considers multiple lipid traits concurrently to delineate the independent effect of each lipid trait on BC susceptibility. Such an approach obviates the need to remove pleiotropic variants and the loss of statistical power that results from

this removal. Therefore, an approach that considers the effects of genetic variants on known risk factors for BC, such as body mass index (BMI) and age at menarche [14–20], could increase power.

While MR assesses evidence for a causal relationship, genome-wide genetic correlation analysis determines whether 2 traits simply have a shared genetic basis. Local genetic correlation analyses test whether 2 traits have shared heritability that is localized to specific genomic regions. These loci may then harbor causal variants and genes that contribute to heritability of both traits. Jiang and colleagues recently estimated genome-wide genetic correlation between lipid traits and BC risk [21]. This study did not find a statistically significant association between any lipid trait and BC using lipid summary statistics, though a previous study with a smaller BC GWAS sample size did report a nominally significant ($P < 0.05$) negative genetic correlation between triglycerides and BC risk [16]. Both these studies used the same method to estimate genome-wide genetic correlations [15], and neither tested for local genetic correlations between lipid traits and BC risk.

In what follows, we apply the causal inference framework of MR to determine whether genetically elevated lipid traits modify BC susceptibility (both all BC and BC stratified by ER status). We take advantage of a recent GWAS for lipid levels performed in up to 215,551 individuals of European ancestry [22], which has not been previously applied to MR studies of BC, to provide power for our causal inference analyses. We utilize several MR techniques, including single-exposure, multivariable, and gene-specific approaches. Of chief concern in modern MR studies, including prior studies of BC and lipids, is confounding due to pleiotropy. For instance, a genetic variant may affect lipid levels indirectly through some other biomarker. If this biomarker directly affects BC risk, this could confound the MR analysis and cause an incorrect inference of a causal effect of lipid levels on BC. Our gene-specific approach utilizes only genetic variants near core HDL pathway genes to minimize this concern. Additionally, we use a multivariable approach that assesses the effects of lipid traits independent of one another and of BMI and age at menarche. Finally, we perform genetic correlation analyses to look for both genome-wide and locus-based correlation in effect sizes between lipids and BC.

## Methods

### Study populations

Lipid GWAS summary statistics were obtained from the Million Veteran Program (MVP) (up to 215,551 European individuals) [22] and the Global Lipids Genetics Consortium (GLGC) (up to 188,577 genotyped individuals) [12]. As additional exposures in multivariable MR analyses, we used BMI summary statistics from a meta-analysis of GWASs in up to 795,640 individuals and age at menarche summary statistics from a meta-analysis of GWASs in up to 329,345 women of European ancestry [17,23]. GWAS summary statistics from 122,977 BC cases and 105,974 controls were obtained from the Breast Cancer Association Consortium (BCAC) [11]. The MVP received ethical and study protocol approval from the Veteran Affair Central Institutional Review Board in accordance with the principles outlined in the Declaration of Helsinki, and written consent was obtained from all participants. For the Willer and colleagues [12] and BCAC [11] data sets, we refer the reader to the primary GWAS manuscripts and their supplementary material for details on consent protocols for each of their respective cohorts. More details on these cohorts are in the S1 Text.

### Lipid meta-analysis

We performed a fixed-effects meta-analysis between each lipid trait (Total cholesterol [TC], LDL, HDL, and triglycerides [TGs]) in GLGC and the corresponding lipid trait in the MVP

cohort [12,22] using the default settings in PLINK [24]. There is some genomic inflation in these meta-analysis association statistics, but linkage disequilibrium (LD)-score regression intercepts demonstrate that this inflation is in large part due to polygenicity and not population stratification (S1 Fig).

## MR analyses

MR analyses were performed using the TwoSampleMR R package version 0.4.13 (https://github.com/MRCIEU/TwoSampleMR) [25]. For all analyses, we used a 2-sample MR framework, with exposure(s) (lipids, BMI, age at menarche) and outcome (BC) genetic associations from separate cohorts. Unless otherwise noted, MR results reported in this manuscript used inverse-variance weighting assuming a multiplicative random effects model. For single-trait MR analyses, we additionally employed Egger regression [26], weighted median [27], and mode-based [28] estimates. SNPs associated with each lipid trait were filtered for genome-wide significance ($P < 5 \times 10^{-8}$) from the MVP lipid study [12], and then we removed SNPs in LD ($r^2 < 0.001$ in UK10K consortium) [29] in order to obtain independent variants. All genetic variants were harmonized using the TwoSampleMR harmonization function with default parameters. Each of these independent, genome-wide significant SNPs was termed a genetic instrument. We estimated that these single-trait MR genetic instruments had 80% power to reject the null hypothesis, with a 1% error rate, for the following odds ratio (OR) increases in BC risk due to a standard deviation increase in lipid levels: HDL, 1.057; LDL, 1.058; TGs, 1.055; TC, 1.060 [30,31]. We tested for directional pleiotropy using the MR-Egger regression test [26]. To reduce heterogeneity in our genetic instruments for single-trait MR, we employed a pruning procedure (S1 Text). Genetic instruments used in single-trait MR are listed in S1 Table. For multivariable MR experiments [32,33], we generated genetic instruments by first filtering the genotyped variants for those present across all data sets. For each trait and data set combination (Yengo and colleagues [23] for BMI; Day and colleagues for age at menarche [17]; MVP and GLGC for HDL, LDL, and TGs), we then filtered for genome-wide significance ($P < 5 \times 10^{-8}$) and for linkage disequilibrium ($r^2 < 0.001$ in UK10K consortium) [29]. We performed tests for instrument strength and validity [34], and each multivariable MR experiment had sufficient instrument strength. We removed variants driving heterogeneity in the ratio of outcome/exposure effects causing instrument invalidity (S1 Text). Genetic instruments used in multivariable MR are listed in S2 Table. Because the MR methods and tests we employed are highly correlated, we did not apply a multiple testing correction to the reported P-values.

## Core HDL and LDL pathway genetic instrument development

We defined sets of core genes for HDL or LDL that met the following criteria: (1) their protein products are known to play a key role in HDL or LDL biology (plus *HMGCR* and *NPC1L1*, 2 targets of LDL-lowering drugs, in the LDL gene set), and (2) there were conditionally independent lipid trait-associated variants within 100 kb upstream or downstream of the RefSeq coordinates for the gene (or locus, in the case of Apolipoprotein E [*APOE*]-Apolipoprotein C [*APOC*]*1-APOC4-APOC2* and Apolipoprotein A [*APOA*]*4-APOC3-APOA1*) [22]. We then used the conditional HDL or LDL association statistics from Klarin and colleagues for those genes in gene-specific MR analyses [22]. The loci included in each set and the genetic instruments used in each locus-specific MR are listed in S3 Table. We performed a separate fixed-effects inverse-variance weighted MR with the conditionally independent genetic instruments at each gene and performed fixed-effects inverse-variance weighted meta-analysis of the results across HDL or LDL genes using the R package meta [35].

## Genetic correlation analyses

We performed cross-trait LD-score regression using the LDSC toolkit, available at https://github.com/bulik/ldsc, with default parameters [15], with the BCAC association statistics for BC and our meta-analysis of GLGC and MVP for lipid associations. In addition, we used the ρ-Hess software both for whole-genome genetic correlation and for local genetic correlation analysis [36], using the UK10K reference panel, and the LD-independent loci published in Berisa and colleagues to partition the genome [37]. We used a Bonferroni significance threshold based on the number of these independent loci (1,704 loci). There was minor cohort overlap between the GLGC and BC GWASs because of the EPIC cohort [10]. We included this overlap when performing ρ-Hess, available at https://huwenboshi.github.io/hess/, using the cross-trait LD-score intercept to estimate phenotypic correlation [36]. The association of the lead BC and lipid SNPs at the ABO locus was obtained using the GTEx v8 data set [38].

## Analysis plan

Our study did not develop a prospective analysis plan. We began by testing for a potential causal relationship between lipids and BC risk using single-trait two-sample MR with lipid genetic associations from the GLGC. After this experiment showed a significant relationship, we tested whether it persisted when we corrected for correlated phenotypes with multivariable MR. Following significant results with the GLGC summary statistics, we decided to confirm these findings using genetic associations from the larger MVP cohort. After our results with MVP confirmed our initial findings, we performed additional MR sensitivity analyses and locus-specific MR. In parallel to our MR experiments, we measured the cross-trait and local genetic correlations between these traits. This study is reported as per the Strengthening the Reporting of Observational Studies in Epidemiology (STROBE) and STROBE-MR guidelines (S1 STROBE Checklist, S1 STROBE-MR Checklist) [39,40].

## Results

### Single-trait MR in BC

We first performed single-trait MR analyses using summary statistics from MVP [22] for each of 4 lipid traits (i.e., TC, HDL-cholesterol, LDL-cholesterol, and TGs) as the intermediate biomarkers and risk for all BCs as the outcome (S2 Fig). We observed a significant relationship between genetically elevated HDL and BC risk (OR = 1.10 per standard deviation of lipid level increase, 95% confidence interval [CI] = 1.04–1.17, P = 2.1 x $10^{-3}$) and genetically decreased TG levels and BC risk (OR = 0.93, 95% CI = 0.88–0.99, P = 0.015; S4 Table). Sensitivity analyses identified heterogeneity (Methods, S5 Table), but there was no evidence of bias from directional pleiotropy (Methods, S6 Table). To mitigate concerns of instrument heterogeneity, we removed variants from our genetic instrument for each lipid trait that were responsible for instrument heterogeneity (S1 Text) and again observed a relationship with HDL-cholesterol (OR = 1.08, 95% CI = 1.04–1.13, P = 7.4 × $10^{-5}$) and TGs (OR = 0.94, 95% CI = 0.90–0.98, P = 2.6 × $10^{-3}$) (Fig 1, S3 and S4 Figs, S7 Table). Because HDL and TGs are inversely correlated [15,41], the opposing relationship between these 2 lipid traits and BC could be expected in single-trait analyses.

To confirm that our results using lipid genetic associations from MVP were not due to heterogeneity between data sets, we also tested the relationship between lipid traits and BC using a meta-analysis of the 2 major lipid GWASs from MVP and GLGC and from GLGC alone. Overall, single-trait MR analyses with the meta-analysis and GLGC lipid associations produced consistent results to those with MVP alone (S5 Fig). In a reciprocal single-trait MR testing the

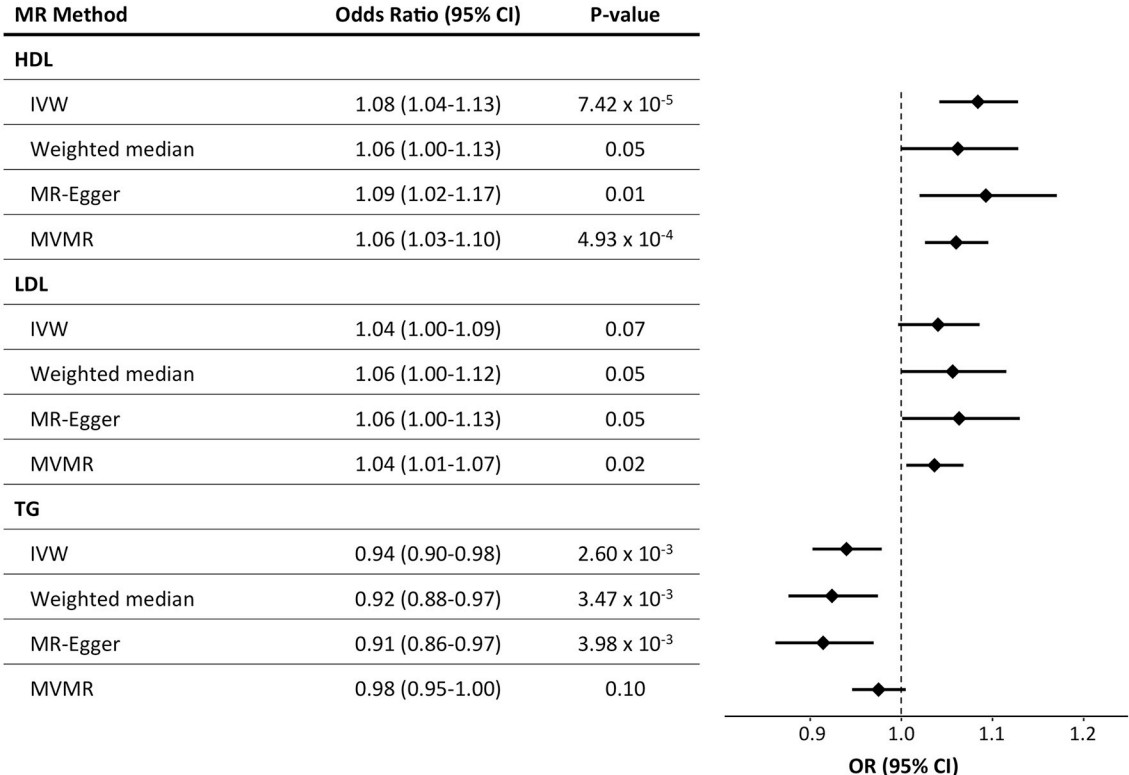

| MR Method | Odds Ratio (95% CI) | P-value |
|---|---|---|
| **HDL** | | |
| IVW | 1.08 (1.04-1.13) | $7.42 \times 10^{-5}$ |
| Weighted median | 1.06 (1.00-1.13) | 0.05 |
| MR-Egger | 1.09 (1.02-1.17) | 0.01 |
| MVMR | 1.06 (1.03-1.10) | $4.93 \times 10^{-4}$ |
| **LDL** | | |
| IVW | 1.04 (1.00-1.09) | 0.07 |
| Weighted median | 1.06 (1.00-1.12) | 0.05 |
| MR-Egger | 1.06 (1.00-1.13) | 0.05 |
| MVMR | 1.04 (1.01-1.07) | 0.02 |
| **TG** | | |
| IVW | 0.94 (0.90-0.98) | $2.60 \times 10^{-3}$ |
| Weighted median | 0.92 (0.88-0.97) | $3.47 \times 10^{-3}$ |
| MR-Egger | 0.91 (0.86-0.97) | $3.98 \times 10^{-3}$ |
| MVMR | 0.98 (0.95-1.00) | 0.10 |

**Fig 1. Results of MR analyses of the effects of lipids on BC risk.** Results plotted are after pruning for instrument heterogeneity. The forest plot on the right displays the OR of the effect of a 1-standard–deviation increase in genetically determined HDL-cholesterol on BC risk as a diamond, with error bars representing the 95% CI. The vertical dotted line delineates an OR of 1, i.e., no effect of the exposure on BC risk. BC, breast cancer; BMI, body mass index; CI, confidence interval; HDL, high-density lipoprotein; IVW, inverse-variance weighted MR; LDL, low-density lipoprotein; MR, Mendelian randomization; MVMR, multivariable MR; OR, odds ratio; TG, triglyceride.

effect of genetically determined BC risk on each lipid trait, we observed no relationship with HDL- or LDL-cholesterol (S8 Table) but did see a relationship with TGs. However, a Steiger test for directionality confirmed that using BC as the outcome was the correct causal direction for all lipid traits (S7 Table) [42]. We also performed genetic instrument pruning in the same manner as Nowak and colleagues: removing genetic instruments for LDL, HDL, and TGs that were associated with at least one of the 2 other lipid traits (P < 0.001) [10]. After this pruning, we did not find a significant relationship with LDL, HDL, or TG, and we note that this pruning procedure resulted in considerably larger CIs spanning OR = 1 for all traits, with reversed direction of effect estimates for LDL and TGs (S9 Table).

## Multivariable MR with age at menarche and BMI as exposures

It has been previously observed that BMI and age at menarche are both genetically corre-lated and epidemiologically associated with both BC [20,43,44] and lipid traits [15,41]. To incorporate these potential confounders into our causal inference framework, we performed multivariable MR analyses using all 3 lipid traits (genetic effect estimates from MVP), age at menarche, and BMI as exposures and BC risk as the outcome (Fig 1). We observed signifi-cant relationships between genetically influenced HDL, LDL, BMI, and age at menarche with BC (HDL: OR = 1.06, 95% CI = 1.03–1.10, P = $4.93 \times 10^{-4}$; LDL: OR = 1.04, 95%

CI = 1.01–1.07, P = 0.02; BMI: OR = 0.90, 95% CI = 0.87–0.94, P = $1.15 \times 10^{-6}$; age at menarche: OR = 0.96, 95% CI = 0.93–0.99, P = $2.44 \times 10^{-3}$), but not TGs (OR = 0.98, 95% CI = 0.95–1.00, P = 0.10) (Fig 1, S10 Table). Our results were consistent before and after pruning for genetic instrument heterogeneity (S10 Table) and when using summary statistics from 3 independent subsets of the BC data set (S6 Fig, S10 Table). We also performed multivariable MR with pairs of lipid traits with genetic effect estimates from different data sets (GLGC or MVP), with and without BMI, and saw consistent results (S11 Table, S7 Fig). Considering the genetic correlation between HDL and TGs, the greater significance of the HDL association compared with the TG association with BC in multivariable analysis, and the consistent relationship between HDL and BC across BC data sets, we focused our further MR analyses on the relationship between HDL-cholesterol and BC, in addition to the previously reported association between LDL and BC [10].

## MR with outcome stratified by ER status

We next performed an MR analysis of the relationship between genetically influenced lipids and BC risk stratified by ER-positive (ER+) or negative (ER−) status. We observed similar effect size estimates of the 4 lipid traits on the BC subtypes as on BC not stratified by subtype (S8 Fig). A formal test for heterogeneity found no evidence to reject the null hypothesis of homogeneity between the cancer subtypes (e.g., HDL: Cochran's Q = $6.6 \times 10^{-5}$, P = 0.99; S12 Table). Thus, we observed no substantive difference in the relationship from any lipid trait to ER+ or ER− BCs, consistent with the strong genetic correlation between these 2 BC subtypes (cross-trait LD-score regression genetic correlation estimate = 0.62, P = $2.9 \times 10^{-83}$). When we used ER+ or ER− BCs as the outcome in multivariable MR, we also saw consistent effects as the analysis with all BCs as the outcome (S13 Table, S9 Fig).

## HDL and LDL pathway gene-specific MR

We next examined associations for BC risk at genetic variants near core HDL or LDL genes. We identified conditionally independent associations at genes that were previously annotated with a core role in the metabolism of each lipid trait or an established drug target (HDL: ATP Binding Cassette Subfamily A Member 1 [*ABCA1*], *APOA4-APOC3-APOA1*, *APOE-APO-C1-APOC4-APOC2*, Cholesteryl Ester Transfer Protein [*CETP*], Lecithin-Cholesterol Acyltransferase [*LCAT*], Lipase C Hepatic Type [*LIPC*], Lipase G Endothelial Type [*LIPG*], Phospholipid Transfer Protein [*PLTP*], Scavenger Receptor Class B Member 1 [*SCARB1*]; LDL: Apolipoprotein B [*APOB*], 3-Hydroxy-3-Methylglutaryl-CoA Reductase [*HMGCR*], LDL Receptor [*LDLR*], Lipoprotein(A) [*LPA*], Myosin Regulatory Light Chain Interacting Protein [*MYLIP*], NPC1-Like Intracellular Cholesterol Transporter 1 [*NPC1L1*], Proprotein Convertase Subtilisin/Kexin Type 9 [*PCSK9*]) (Methods, S3 Table). For each gene or locus with at least 2 conditionally independent genetic instruments (all except *LCAT* and *MYLIP*), we performed inverse-variance-weighted MR (fixed-effects model) with conditional HDL or LDL effect size estimates as the exposure and BC risk as the outcome (S10 and S11 Figs). We observed significant (P < 0.05) positive relationships between HDL and BC risk at 3 loci (*ABCA1*, *APOE-APOC1-APOC4-APOC2*, and *CETP*; Fig 2), and between LDL and BC risk at 1 locus (*HMGCR*, S12 Fig). Combining the effect estimates across core genes in a meta-analysis, we observed a significant positive relationship for HDL (OR = 1.11, 95% CI = 1.06–1.16, P < 0.001; Fig 2) and LDL (OR = 1.07, 95% CI = 1.01–1.14, P = 0.02; S12 Fig). There was no evidence of heterogeneity across loci in either meta-analysis (HDL: Q = 6.63, P = 0.47; LDL: Q = 5.53, P = 0.35).

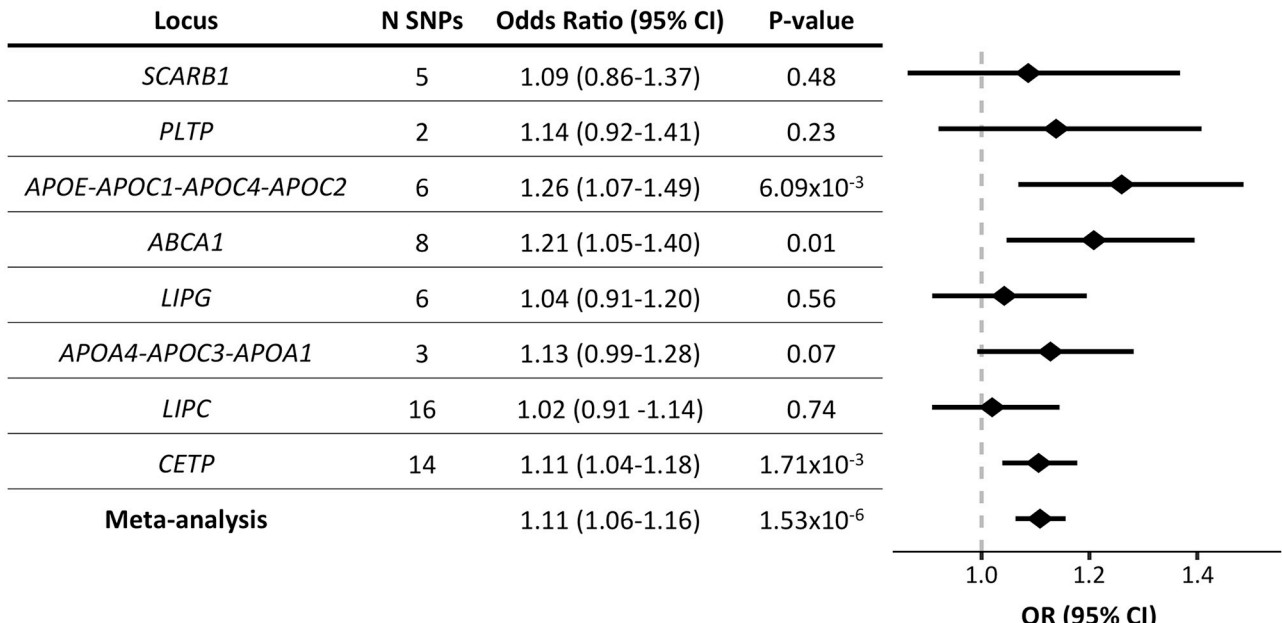

| Locus | N SNPs | Odds Ratio (95% CI) | P-value |
|---|---|---|---|
| *SCARB1* | 5 | 1.09 (0.86-1.37) | 0.48 |
| *PLTP* | 2 | 1.14 (0.92-1.41) | 0.23 |
| *APOE-APOC1-APOC4-APOC2* | 6 | 1.26 (1.07-1.49) | 6.09x10⁻³ |
| *ABCA1* | 8 | 1.21 (1.05-1.40) | 0.01 |
| *LIPG* | 6 | 1.04 (0.91-1.20) | 0.56 |
| *APOA4-APOC3-APOA1* | 3 | 1.13 (0.99-1.28) | 0.07 |
| *LIPC* | 16 | 1.02 (0.91 -1.14) | 0.74 |
| *CETP* | 14 | 1.11 (1.04-1.18) | 1.71x10⁻³ |
| **Meta-analysis** | | 1.11 (1.06-1.16) | 1.53x10⁻⁶ |

**Fig 2. MR results for HDL gene-specific instruments and meta-analysis of effect estimates across genes.** The forest plot on the right displays the OR of the effect of a 1-standard–deviation increase in genetically determined HDL-cholesterol for each locus on BC risk as a diamond, and the error bars represent the 95% CI of the effect estimate. The vertical dotted line delineates an OR of 1, i.e., no effect of the exposure on BC risk. For HDL gene-specific instruments, see S3 Table. *ABCA1*, ATP Binding Cassette Subfamily A Member 1; *APOC*, Apolipoprotein C; *APOE*, Apolipoprotein E; BC, breast cancer; *CETP*, Cholesteryl Ester Transfer Protein; CI, confidence interval; HDL, high-density lipoprotein; *LIPC*, Lipase C, Hepatic Type; LIP*G*, Lipase G, Endothelial Type; MR, Mendelian randomization; N SNPs, number of genetic instruments included in each locus's MR analysis; OR, odds ratio; *PLTP*, Phospholipid Transfer Protein; *SCARB1*, Scavenger Receptor Class B Member 1.

## Genome-wide and local genetic correlation

If cholesterol levels were a causal risk factor for BC, we might expect a correlation between the strength of genetic association with these 2 traits at genetic variants across the entire genome in addition to those at genome-wide significant loci. To answer this question, we utilized 2 approaches to estimate genetic correlation between BC and lipid traits. Using the ρ-Hess method, we found significant (P < 0.05) correlations between LDL (P < 0.001) and TC (P = 0.01) and BC, with directions consistent with our MR results (S14 Table). Cross-trait LD-score regression found positive genetic correlation estimates for TC, LDL, and HDL and a negative estimate for TGs (S13 Fig, S14 Table), consistent with our MR and ρ-Hess results [15]. However, the only significant association (P < 0.05) was with TC and ER-negative BC (P = 0.04).

To discover new loci that are enriched for genetic correlation between BC and lipids, we used the ρ-Hess method, which detects genomic regions harboring high genetic correlation between 2 traits [36]. ρ-Hess identified one region that surpassed Bonferroni test correction, with a positive correlation between both LDL and TC and BC (S15 Table). In this region, there are 2 SNPs in high LD (rs532436 and rs635634, r² = 0.99) that are genome-wide significantly associated with LDL (rs532436: P < 0.001), TC (P < 0.001), and BC (P < 0.001). These SNPs lie within an intron of the ABO blood group determining *ABO* gene. rs635634 moderately tags an SNP associated with ABO blood type [41]. However, this SNP is also associated with a change in gene expression of *ABO* in multiple tissues (P < 0.001 in breast mammary tissue) [38].

## Discussion

Using MR, we provide evidence that genetically elevated HDL and LDL levels are associated with increased risk for BC, supporting a causal hypothesis. Previous meta-analyses of observational studies of BC risk and lipids reported a negative association with HDL and no relationship with LDL [4,5], whereas individual studies have reported a positive relationship with HDL [6] or no relationship with HDL or LDL [45,46]. Our analyses help clarify these mixed results and infer a direction of effect, which is not possible in observational studies because of potential reverse causation. Furthermore, we find evidence of genome-wide genetic correlation between some lipid traits and BC and local genetic correlation at the ABO locus. Although some studies have found an association between blood group and BC risk [47], haplotype patterns indicate that ABO gene expression, not blood group, may be the causal mechanism [37]. However, because of the pleiotropic nature of the ABO locus, it is unclear whether the BC association is caused by the lipid associations [41].

Although Nowak and colleagues previously used MR to discover associations between lipids and BC [10], our report presents a thorough reconsideration of these effects. Even after conditioning on the effects of HDL, BMI, and age at menarche, our MR analysis suggests a potential causal relationship between LDL and BC. Nowak and colleagues only found a relationship between HDL-cholesterol and ER+ BC, whereas we found a relationship between HDL-cholesterol and risk for all BCs. We also find a previously unreported association with TGs and BC, though our multivariable analysis suggests this may be explained by correlation between TGs and HDL and not an independent TG effect. In their analyses, Nowak and colleagues used a strict pruning procedure to isolate the effects of each lipid trait. However, this approach reduces power because of the high genetic correlation of these traits. The multivariable approach taken here is an alternative way to estimate the effect of an exposure while accounting for correlated exposures.

Our results largely agree with those reported in the recent MR study of BC and lipids by Beeghly-Fadiel and colleagues [48], published while this manuscript was under peer review. Both studies use multiple types of MR analyses, including approaches accounting for confounding by BMI and age at menarche, and report a positive association between HDL and BC and a negative association between TGs and BC. However, our report provides complementary analysis and data that support the central findings of both pieces of work. First, we took advantage of the recently reported MVP lipids GWAS [22], providing a larger number of genetic instruments for all lipid traits considered. Second, we explicitly considered age of menarche and BMI in multivariable models with all lipid traits. It is crucial to consider the collection of each of these risk factors together to estimate a causal effect estimate that is independent of these confounders, as well as across cancer risk strata (i.e., ER status). While Beeghly-Fadiel and colleagues took advantage of access to individual-level data to adjust for confounding factors in their single lipid MR, these confounders were not corrected for in their multivariable MR. Third, while both studies are consistent in their relationship between HDL and BC, we reported a nominal association ($P < 0.05$) with LDL levels when considering all risk and confounding factors jointly. Finally, we present unique, locus-specific MR analyses to show that conditionally independent associations at single loci implicated in HDL or LDL biology are significantly associated with BC risk.

A challenge of MR analyses that use hundreds of genetic instruments is that we do not know the mechanism of action of these instruments on the exposure trait. By focusing on loci with mechanistic evidence of a direct effect on lipid levels, we can remove uncertainty about potential pleiotropic effects on BC risk. Thus, the significant relationships observed in our locus-specific MR analyses across HDL or LDL pathway genes provide additional evidence for

a direct effect of increased HDL or LDL levels on BC risk. Furthermore, these genes represent potential or established drug targets, and each locus-specific MR experiment provides preliminary evidence for the therapeutic potential of cholesterol modification on BC prevention.

Substantial effort has been spent developing HDL-raising therapies for cardiovascular disease prevention; however, recent studies have proposed an increase in all-cause mortality in individuals with high HDL levels [49,50]. Our results suggest that therapies that aim to reduce cardiovascular risk by raising HDL levels might have an unintended consequence of elevated BC risk. Specifically, our gene-based score using HDL-raising variation at the *CETP* locus predicted that *CETP*-based inhibition would elevate BC risk (OR = 1.11, 95% CI = 1.04–1.18, P < 0.001). Additionally, 2 recent MR studies reported causal evidence between elevated HDL and risk for age-related macular degeneration [51,52]. These potential disease-increasing consequences may not have been possible to identify in safety trials, given the limited window of study to monitor progression or incidence of disease, the putative causal effect estimates, and the demographics of the study population (i.e., a higher proportion of male participants) [53]. Our result supports the use of human genetics data as both a novel strategy for therapeutic targeting and for the discovery of potential drug complications to direct long-term post-clinical–trial follow-up [54].

We note several caveats to our analyses. The first is that MR makes several assumptions that must be met for accurate causal inference [55,56]. Although we used statistical methods that try to detect and correct for violations of these assumptions, these methods are not guaranteed to correct for all types of confounding, and alternative causal inference frameworks outside of MR are warranted. Secondly, MR is only able to make inferences about trait associations in the populations from which the GWASs are derived. We are unaware of evidence that BC or lipid genetic architecture varies significantly across populations, but if this was the case, our findings may not be generalizable to these different scenarios. However, the concordance between our results using the MVP and GLGC GWASs mitigate this concern with regards to potential heterogeneity in lipid genetic architecture. Thirdly, the estimated lipid/BC effect sizes represent only the population-averaged causal effect and may not generalize well to other populations or settings [57]. We note that our effect estimates may be attenuated because of association of lipid instrumental variables with the use of lipid-lowering medication, and that we cannot be certain that the true underlying causal exposure is lipid levels rather than another phenotype for which lipids are a proxy. However, we are not aware of any process for which lipids is a proxy through which BC would be affected, and our gene-specific approach minimizes this concern. Additionally, it is perhaps surprising that we did not find a significant genetic correlation between BC and lipids using cross-trait LD-score regression; however, our result corroborates a previous study that performed this analysis using smaller GWASs [16]. Our lack of significant results could be caused by limited polygenicity of either trait, which decreases the power of this method [15]. We do find significant cross-trait heritability between BC and 2 lipid traits (TC and LDL) using the ρ-Hess method. The discrepancy between LDSC and ρ-Hess may be due to a difference in the statistical power of these methods that has been previously reported [36].

The analyses presented here do not bring evidence on a specific mechanism for tumorigenesis, but they do bring renewed attention to potential mechanisms requiring future functional study. Cholesterol and its oxysterol metabolites, either in the circulatory system or in the local mammary microenvironment, may have direct effects on mammary tissue growth induction of breast tumorigenesis [58,59].

These findings support a causal relationship between increased HDL-cholesterol and increased BC risk, and this hypothesis warrants further exploration. Statins are widely prescribed to decrease LDL levels; however, statins also increase HDL levels. If further research

substantiates the relationship between higher HDL levels and increased BC risk, the consensus that HDL is "good cholesterol," or of benign effect, may require re-evaluation.

## Supporting information

**S1 STROBE checklist. Reporting document following the STROBE guidelines for our study.** STROBE, Strengthening the Reporting of Observational Studies in Epidemiology. (DOCX)

**S1 STROBE-MR checklist. Reporting document following the preliminary STROBE-MR guidelines for our study.** MR, Mendelian randomization; STROBE, Strengthening the Reporting of Observational Studies in Epidemiology. (DOCX)

**S1 Fig. QQ plots for lipid association statistics.** Generated from a meta-analysis of Klarin and colleagues [22] and Willer and colleagues [12]. LD-score regression intercepts and standard errors from the meta-analysis association statistics were as follows: TC, 1.1293 (0.1143); TG, 1.0317 (0.0656); LDL, 1.0933 (0.1001); HDL, 1.1715 (0.0758); see also S14 Table. HDL, high-density lipoprotein; LD, linkage disequilibrium; LDL, low-density lipoprotein; TC, total cholesterol; TG, triglyceride; λGC, genomic inflation factor. (PDF)

**S2 Fig. Scatter plots of unpruned single-trait MR genetic instruments' effect estimates on exposure and outcome.** Plotted are the genetic instruments included in unpruned single-trait MR analyses. Each plot contains effect estimates from MVP for one of 4 lipid traits (HDL, LDL, TC, TGs) on the x-axis and effect estimates for risk of all BCs (allBC) on the y-axis. Error bars represent the 95% CI, and regression lines represent the slope estimate from one of 3 MR tests: IVW (light blue), Egger regression (dark blue), and weighted median (green). BC, breast cancer; CI, confidence interval; HDL, high-density lipoprotein; IVW, inverse-variance weighted; LDL, low-density lipoprotein; MR, Mendelian randomization; TC, total cholesterol; TG, triglyceride. (PDF)

**S3 Fig. Scatter plots of pruned single-trait MR genetic instruments' effect estimates on exposure and outcome.** Scatter plots of genetic instruments included in pruned single-trait MR analyses. Genetic instruments were pruned to pass heterogeneity test. Each plot contains effect estimates from MVP for one of 4 lipid traits (HDL, LDL, TC, TG) on the x-axis and effect estimates for risk of all BCs on the y-axis. Error bars represent the 95% CI, and regression lines represent the slope estimate from one of 3 MR tests: IVW (light blue), Egger regression (dark blue), and weighted median (green). BC, breast cancer; CI, confidence interval; HDL, high-density lipoprotein; IVW, inverse-variance weighted; LDL, low-density lipoprotein; MR, Mendelian randomization; TC, total cholesterol; TG, triglyceride. (PDF)

**S4 Fig. Results of single-trait MR analyses with a lipid trait as the exposure and risk for all BCs as the outcome.** The exposure association data for this analysis were from MVP. Genetic instruments were pruned to pass heterogeneity test. Error bars represent the 95% CI. Estimates were calculated using the IVW method. *P < 0.05; **P < 0.001. See S7 Table for ORs, CIs, and P-values. BC, breast cancer; CI, confidence interval; HDL, high-density lipoprotein; IVW, inverse-variance weighted; LDL, low-density lipoprotein; MR, Mendelian randomization; MVP, Million Veteran Program; OR, odds ratio; TC, total cholesterol; TG, triglyceride. (PDF)

**S5 Fig. Single-trait MR with lipid association statistics from MVP, GLGC, or MVP + GLGC meta-analysis.** Genetic association statistics for all BCs were used for the outcome. Genetic instruments were pruned to pass heterogeneity test. Error bars represent the 95% CI. Estimates were calculated using the IVW method. BC, breast cancer; CI, confidence interval; GLGC, Global Lipids Genetics Consortium; HDL, high-density lipoprotein; IVW, inverse-variance weighted; LDL, low-density lipoprotein; MR, Mendelian randomization; MVP, Million Veteran Program; TC, total cholesterol; TG, triglyceride.
(PDF)

**S6 Fig. Multivariable MR analyses stratified by 3 independent subsets of the BCAC data set.** Results of multivariable MR analyses with 3 lipid traits (HDL, LDL, TG), BMI, and AaM as exposures and BC risk as the outcome. Each panel presents multivariable MR results using BC summary statistics from an independent subset of the BCAC data set (Oncoarray, iCOGS, or GWAS) or from the meta-analysis of all 3 together (BC meta-analysis, S1 Text). Results plotted are after pruning for instrument heterogeneity. Error bars represent the 95% CI. Estimates were calculated using the IVW method. *P < 0.05; **P < 0.001. See S10 Table for ORs, CIs, and P-values. AaM, age at menarche; BC, breast cancer; BCAC, Breast Cancer Association Consortium; BMI, body mass index; CI, confidence interval; GWAS, genome-wide association study; HDL, high-density lipoprotein; IVW, inverse-variance weighted; LDL, low-density lipoprotein; MR, Mendelian randomization; OR, odds ratio; TG, triglyceride.
(PDF)

**S7 Fig. Multivariable MR analyses using lipid genetic associations from either MVP or GLGC.** Results of multivariable MR analyses including 2 lipid traits as exposures: (A) LDL and HDL or (B) TGs and HDL, with and without BMI as an additional exposure and with risk for all BCs as the outcome. Results plotted are after pruning for instrument heterogeneity. The lipid effect estimates were from one of 2 GWAS data sets (MVP or GLGC), and the results of each combination of lipid data sets are in a single plot. Error bars represent the 95% CI. Estimates were calculated using the IVW method. *P < 0.05; **P < 0.001. See S11 Table for ORs, CIs, and P-values. BC, breast cancer; BMI, body mass index; CI, confidence interval; GLGC, Global Lipids Genetics Consortium; GWAS, genome-wide association study; HDL, high-density lipoprotein; IVW, inverse-variance weighted; LDL, low-density lipoprotein; MR, Mendelian randomization; MVP, Million Veteran Program; OR, odds ratio; TG, triglyceride.
(PDF)

**S8 Fig. Single-trait MR with BC outcomes stratified by ER subtypes.** Results of single-trait MR with each lipid trait as an exposure, and one of 3 BC traits as the outcome: all BC, ER − BCs only, or ER+ BCs only. Error bars represent the 95% CI. Estimates were calculated using a fixed-effects IVW method after pruning for instrument heterogeneity. Lipid association statistics come from the MVP data. **P < 0.001, *P < 0.05. See S12 Table for ORs, CIs, and P-values. BC, breast cancer; CI, confidence interval; ER, estrogen receptor; HDL, high-density lipoprotein; IVW, inverse-variance weighted; LDL, low-density lipoprotein; MR, Mendelian randomization; OR, odds ratio; TC, total cholesterol; TG, triglyceride.
(PDF)

**S9 Fig. Multivariable MR analyses with BC outcomes stratified by ER subtypes.** Results of multivariable MR analyses with 3 lipid traits (HDL, LDL, TGs), BMI, and AaM as exposures; and all BCs, ER−, or ER+ BC as the outcome. Results plotted are after pruning for instrument heterogeneity. Error bars represent the 95% CI. Estimates were calculated using the IVW method. *P < 0.05; **P < 0.001. See S13 Table for ORs, CIs, and P-values. AaM, age at menarche; BC, breast cancer; BMI, body mass index; CI, confidence interval; ER, estrogen receptor;

HDL, high-density lipoprotein; IVW, inverse-variance weighted; LDL, low-density lipoprotein; MR, Mendelian randomization; TG, triglyceride.
(PDF)

**S10 Fig. Genetic instruments' effect estimates on HDL and BC at each canonical HDL metabolism pathway loci.** Conditionally independent HDL-associated SNPs at canonical HDL metabolism pathway genes, plotted by their conditional effect estimates on HDL (from MVP) and effect estimates on all BCs. Error bars represent 95% CIs. The dashed green line represents the regression line from fixed-effects IVW MR. BC, breast cancer; CI, confidence interval; HDL, high-density lipoprotein; IVW, inverse-variance weighted; MR, Mendelian randomization; MVP, Million Veteran Program.
(PDF)

**S11 Fig. Genetic instruments' effect estimates on LDL and BC at each canonical LDL metabolism pathway loci.** Conditionally independent LDL-associated SNPs at canonical LDL metabolism pathway genes, plotted by their conditional effect estimates on LDL (from MVP) and effect estimates on all BCs. Error bars represent 95% CIs. The dashed green line represents the regression line from fixed-effects IVW MR. BC, breast cancer; CI, confidence interval; IVW, inverse-variance weighted; LDL, low-density lipoprotein; MR, Mendelian randomization; MVP, Million Veteran Program.
(PDF)

**S12 Fig. LDL locus-specific MR results with LDL as exposure and BC risk as outcome.** Forest plot of MR results for LDL gene-specific instruments (see S4 Fig) and meta-analysis of effect estimates across genes. Estimates were calculated using a fixed-effects IVW method. BC, breast cancer; CI, confidence interval; IVW, inverse-variance weighted; LDL, low-density lipoprotein; MR, Mendelian randomization; N SNPs, number of genetic instruments included in MR; OR, odds ratio.
(PDF)

**S13 Fig. Genetic correlations between lipid and BC traits.** Results of LD-score regression testing for genetic correlation between each lipid trait and 3 BC traits: all BC, ER– BCs only, or ER+ BCs only. Error bars represent the 95% CI. Lipid association statistics were from a meta-analysis of GLGC and MVP. BC, breast cancer; CI, confidence interval; ER, estrogen receptor; GLGC, Global Lipids Genetics Consortium; HDL, high-density lipoprotein; LD, linkage disequilibrium; LDL, low-density lipoprotein; MVP, Million Veteran Program; TC; total cholesterol; TG, triglyceride.
(PDF)

**S1 Table. Summary statistics for genetic instruments used in single-trait MR analyses.** Lipid exposure summary statistics are from the MVP European data set and BC summary statistics from the BCAC consortium meta-analysis. SNP: rsID of genetic instrument; exposure: lipid trait for exposure statistics; effect_allele.exposure: allele used for lipid and BC effect estimates; other_allele.exposure: noneffect allele; beta.exposure: effect size estimate for lipid trait; se.exposure: standard error of lipid effect size estimate; pval.exposure: P-value of lipid trait effect estimate; beta.bc: effect size estimate of BC risk; se.bc: standard error of BC risk effect estimate; pval.bc: P-value for BC risk effect estimate; inclPruned: logical, was SNP included in pruned single-trait MR analysis. BC, breast cancer; BCAC, Breast Cancer Association Consortium; HDL, high-density lipoprotein; LDL, low-density lipoprotein; MR, Mendelian randomization; MVP, Million Veteran Program; TC, total cholesterol; TG, triglyceride.
(XLSX)

**S2 Table. Summary statistics for genetic instruments used in multivariable MR analyses.**
SNP: rsID of genetic instrument; expZ: trait and data set used as first/second/. . . exposure
(e.g., if exp1 is ldl_mvp, LDL summary statistics from GLGC were used as the first exposure);
expZ_beta: effect size estimate for trait expZ; expZ_se: standard error of effect size estimate for
trait expZ; expZ_pval: P-value of effect size estimate for trait expZ; bc_beta: BC risk effect size
estimate; bc_se: standard error of BC effect size estimate; bc_pval: P-value of BC effect size esti-
mate; test: unique identifier for each 2, 3, or 5-exposure MVMR experiment included in this
table. AaM, age at menarche; BC, breast cancer; BMI, body mass index; GLGC, Global Lipids
Genetics Consortium; HDL, high-density lipoprotein; LDL, low-density lipoprotein; MR,
Mendelian randomization; MVMR, multivariable MR; MVP, Million Veteran Program; TC;
total cholesterol; TG, triglyceride.
(XLSX)

**S3 Table. Conditionally independent summary statistics for HDL or LDL associations
used in locus-specific MR analyses.** Data are from the conditional analysis of summary
statistics from MVP and GLGC meta-analysis published in Klarin and colleagues [22]. CHR:
chromosome; POS: base position (HG19); SNP: rsID; effect.allele: allele used for effect size esti-
mate; other.allele: noneffect allele; effect.allele.freq: frequency of effect allele; conditional.beta:
effect size estimate from conditional analysis; conditional.se: standard error of conditional
effect size estimate; conditional.p: P-value of conditional effect size estimate; Locus: the
HDL or LDL gene or locus for MR analysis; Trait: lipid trait used as exposure (HDL or LDL).
GLGC, Global Lipids Genetics Consortium; HDL, high-density lipoprotein; LDL, low-density
lipoprotein; MR, Mendelian randomization; MVP, Million Veteran Program.
(XLSX)

**S4 Table. Single-trait MR results with unpruned lipid traits as the exposure and all BCs as
the outcome for a range of MR methods.** Lipid association statistics are from MVP. Expo-
sure: lipid trait used as the exposure; Method: MR method used; N SNPs: number of genetic
instruments included in analysis; CI_95_L: lower bound of 95% CI; CI_95_U: upper bound of
95% CI; P: P-value of MR test. BC, breast cancer; CI, confidence interval; HDL, high-density
lipoprotein; LDL, low-density lipoprotein; MR, Mendelian randomization; MVP, Million Vet-
eran Program; OR, odds ratio; TC, total cholesterol; TG, triglyceride.
(XLSX)

**S5 Table. Heterogeneity analyses by Cochran's Q of unpruned single-trait MR.** Lipid asso-
ciation statistics are from MVP. Estimates are from the IVW method, and the outcome trait
was risk for all BC. Exposure: lipid trait used as the exposure; Q: Cochran's Q statistic; Q_df:
degrees of freedom in Cochran's Q test; Q_P: P-value of Cochran's Q test. BC, breast cancer;
HDL, high-density lipoprotein; IVW, inverse-variance weighted; LDL, low-density lipopro-
tein; MR, Mendelian randomization; MVP, Million Veteran Program; TC, total cholesterol;
TG, triglyceride.
(XLSX)

**S6 Table. Pleiotropy analysis using Egger regression of unpruned single-trait MR.** Lipid
association statistics are from MVP, and the outcome trait was risk for all BC. Exposure: lipid
trait used as the exposure; Egger intercept: intercept estimate from Egger regression; SE: stan-
dard error of intercept estimate; P: P-value of intercept estimate. BC, breast cancer; HDL,
high-density lipoprotein; LDL, low-density lipoprotein; MR, Mendelian randomization; MVP,
Million Veteran Program; TC, total cholesterol; TG, triglyceride.
(XLSX)

**S7 Table. Results of single-trait MR, heterogeneity analyses, and directionality analyses with pruned lipid IV sets.** Lipid summary statistics were from MVP, and MR tests used the IVW method. Heterogeneity analyses used Cochran's Q, and directionality analyses used the Steiger test. Exposure: lipid trait used as the exposure; N_SNPs: number of genetic instruments included in analysis; OR: OR of MR test; CI_95_L: lower bound of 95% CI; CI_95_U: upper bound of 95% CI; MR_P: P-value of MR test; Q: Cochran's Q statistic; Q_df: degrees of freedom in Cochran's Q test; Q_Pval: P-value of Cochran's Q test; SNP_r2_exposure: estimated variance in lipid trait explained by genetic instruments; SNP_r2_outcome: estimated variance in breast cancer risk explained by genetic instruments; Steiger_pval: P-value of Steiger test inference of causal direction; correct_causal_direction: logical, is the causal direction inferred by the Steiger test in the correct direction. CI, confidence interval; HDL, high-density lipoprotein; IV, instrumental variable; IVW, inverse-variance weighted; LDL, low-density lipoprotein; MR, Mendelian randomization; MVP, Million Veteran Program; OR, odds ratio; TC, total cholesterol; TG, triglyceride.
(XLSX)

**S8 Table. Results of a reciprocal single-trait MR testing the effects of BC as the exposure on each lipid trait as the outcome.** Lipid summary statistics were from MVP, and MR tests used the IVW method. Exposure: BC used as exposure for all tests; Outcome: lipid trait used as the outcome; N_SNPs: number of genetic instruments used in MR test; CI_95_L: lower bound of 95% CI; CI_95_U: upper bound of 95% CI; P: P-value of MR test. BC, breast cancer; CI, confidence interval; HDL, high-density lipoprotein; IVW, inverse-variance weighted; LDL, low-density lipoprotein; MR, Mendelian randomization; MVP, Million Veteran Program; OR, odds ratio; TG, triglyceride.
(XLSX)

**S9 Table. Results of single-trait MR analyses after pruning for genetic instruments associated with other lipid traits.** For each exposure, genetic instruments that were associated (P < 0.001) with the other 2 listed lipid traits were removed before this MR analysis. Lipid summary statistics were from MVP, MR tests used the IVW method, and risk for all BCs was the outcome. Exposure: lipid trait used as the exposure; N_SNPs: number of genetic instruments used in MR test; CI_95_L: lower bound of 95% CI; CI_95_U: upper bound of 95% CI; P: P-value of MR test. BC, breast cancer; CI, confidence interval; HDL, high-density lipoprotein; IVW, inverse-variance weighted; LDL, low-density lipoprotein; MR, Mendelian randomization; MVP, Million Veteran Program; OR, odds ratio; TG, triglyceride.
(XLSX)

**S10 Table. Results of multivariable MR with 3 lipid traits, BMI, and AaM, as exposures and BC risk as outcome.** Results are from 4 separate multivariable MR experiments: 3 with outcome summary statistics from independent subsets of the BC data set (BC_Onco, Oncoarray; BC_iCoGS, iCOGS; or BC_GWAS, GWAS), or using the BCAC meta-analysis summary statistics (BC_Meta). Lipid summary statistics are from MVP. Before/after pruning: results from MR before or after pruning for instrument heterogeneity; Exposure: trait used as exposure; Outcome: BC summary statistics used for the outcome; N_SNPs: number of genetic instruments used in MR test; CI_95_L: lower bound of 95% CI; CI_95_U: upper bound of 95% CI; P: P-value of MR test. AaM, age at menarche; BC, breast cancer; BCAC, Breast Cancer Association Consortium; BMI, body mass index; CI, confidence interval; GWAS, genome-wide association study; HDL, high-density lipoprotein; LDL, low-density lipoprotein; MR, Mendelian randomization; MVP, Million Veteran Program; TG, triglyceride.
(XLSX)

**S11 Table. Results of multivariable MR with lipid trait summary statistics from distinct cohorts.** Risk for all BCs was used as the outcome. Within a test, the effect estimates for each lipid trait are from different data sets (MVP or GLGC). Each test is separated by an empty row. Exposure_Dataset: lipid trait (or BMI) used as exposure and the data set for lipid summary statistics; N_SNPs: number of genetic instruments used in MR test; CI_95_L: lower bound of 95% CI; CI_95_U: upper bound of 95% CI; P: P-value of MR test; Before/after pruning: results from MR before or after pruning for instrument heterogeneity. BC, breast cancer; BMI, body mass index; CI, confidence interval; GLGC, Global Lipids Genetics Consortium; HDL, high-density lipoprotein; LDL, low-density lipoprotein; MR, Mendelian randomization; MVP, Million Veteran Program; OR, odds ratio; TG, triglyceride.
(XLSX)

**S12 Table. Results of Cochran's Q test for heterogeneity on single-trait IVW MR results for the effect of a lipid trait on ER+ versus ER− BC.** Trait: lipid trait used as the exposure; ERposBeta: MR effect estimate for ER+ BCs; ERnegBeta: MR effect estimate for ER− BCs; ERposSE: standard error of MR effect estimate for ER+ BCs; ERnegSE: standard error of MR effect estimate for ER− BCs; Q: Cochran's Q test statistic; P: P-value of Cochran's Q test. BC, breast cancer; ER, estrogen receptor; HDL, high-density lipoprotein; IVW, inverse-variance weighted; LDL, low-density lipoprotein; MR, Mendelian randomization; TC, total cholesterol; TG, triglyceride.
(XLSX)

**S13 Table. Results of multivariable MR with ER+ or ER− BC as the outcome.** For each test, 5 traits were used as the exposure traits and ER+ or ER− BC as the outcome. Results are after genetic instrument pruning. Lipid summary statistics are from MVP. N_SNPs: number of genetic instruments used in MR test; CI_95_L: lower bound of 95% CI; CI_95_U: upper bound of 95% CI; P: P-value of MR test. BC, breast cancer; BMI, body mass index; CI, confidence interval; ER, estrogen receptor; HDL, high-density lipoprotein; GLGC, Global Lipids Genetics Consortium; LDL, low-density lipoprotein; MR, Mendelian randomization; MVP, Million Veteran Program; OR, odds ratio; TG, triglycerides.
(XLSX)

**S14 Table. Genome-wide genetic correlation estimates.** Genetic correlation estimates between each lipid trait and risk of all BCs, calculated by 2 methods. Lipid Trait: lipid trait used, Method: method used to estimate genetic correlation; Lipid intercept: estimate of lipid intercept from LDSC; Lipid intercept se: standard error of lipid intercept estimate from LDSC; Cross-trait intercept with BC-all: estimate of cross-trait intercept with risk of all BCs from LDSC; Cross-trait intercept se with BC-all: standard error of estimate of cross-trait intercept with risk of all BCs from LDSC; Genetic covariance: estimate of genetic covariance between lipid trait and BC; Covariance SE: standard error of genetic covariance estimate; Genetic correlation: estimate of genetic correlation between lipid trait and BC; Correlation SE: standard error of estimate of genetic correlation; Correlation Z-score: standard normalized estimate of genetic correlation; Correlation p-value: P-value of genetic correlation. BC, breast cancer; HDL, high-density lipoprotein; Hess, ρ-Hess method; LDL, low-density lipoprotein; LDSC, linkage disequilibrium-score regression; TC, total cholesterol; TG, triglycerides.
(XLSX)

**S15 Table. Local genetic correlations calculated with the ρ-Hess method.** Genomic regions with significant local genetic correlation between BC and lipids using the ρ-Hess method. Only loci that passed Bonferroni correction ($P = 0.05/1{,}703$ partitions $= 2.9 \times 10^{-5}$) are shown. Lipid: lipid trait used; Position: genomic coordinates of loci tested (HG19); N_SNPs: number

of SNPs in partition; K: number eigenvectors used; Local-rhog: local genetic correlation estimate; Variance: variance estimate; SE: standard error of genetic correlation estimate; Z: Z-score of genetic correlation estimate; P: P-value of genetic correlation estimate. BC, breast cancer.

(XLSX)

**S1 Text. Supplementary Text.** Included are details about the GWASs utilized, heterogeneity analysis for single-trait MR, instrument strength and validity assessment for multivariable MR, and a list of investigators associated with the VA MVP (banner author). GWAS, genome-wide association study; MR, Mendelian randomization; MVP, Million Veteran Program; VA, Veterans Affairs.

(DOCX)

## Author Contributions

**Conceptualization:** Kelsey E. Johnson, Katherine M. Siewert, Kara N. Maxwell, Benjamin F. Voight.

**Data curation:** Kelsey E. Johnson, Katherine M. Siewert, Derek Klarin, Scott M. Damrauer, Kyong-Mi Chang, Philip S. Tsao, Themistocles L. Assimes, Benjamin F. Voight.

**Formal analysis:** Kelsey E. Johnson, Katherine M. Siewert, Benjamin F. Voight.

**Funding acquisition:** Kara N. Maxwell, Benjamin F. Voight.

**Investigation:** Kelsey E. Johnson, Katherine M. Siewert, Benjamin F. Voight.

**Methodology:** Kelsey E. Johnson, Katherine M. Siewert, Benjamin F. Voight.

**Project administration:** Kara N. Maxwell, Benjamin F. Voight.

**Resources:** Benjamin F. Voight.

**Supervision:** Kara N. Maxwell, Benjamin F. Voight.

**Visualization:** Kelsey E. Johnson, Benjamin F. Voight.

**Writing – original draft:** Kelsey E. Johnson, Katherine M. Siewert, Kara N. Maxwell, Benjamin F. Voight.

**Writing – review & editing:** Kelsey E. Johnson, Katherine M. Siewert, Derek Klarin, Scott M. Damrauer, Kyong-Mi Chang, Philip S. Tsao, Themistocles L. Assimes, Kara N. Maxwell, Benjamin F. Voight.

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
