## [Decision Letter · Decision Letter 0]

24 Mar 2020

Dear Dr. Siewert,

Thank you very much for submitting your manuscript "Assessing a causal relationship between circulating lipids and breast cancer risk: Mendelian randomization study" (PMEDICINE-D-19-03886) for consideration at PLOS Medicine. 

Your paper was evaluated by a senior editor and discussed among all the editors here. It was also sent to five independent reviewers. The reviews are appended at the bottom of this email and any accompanying reviewer attachments can be seen via the link below:

[LINK]

In light of these reviews, I am afraid that we will not be able to accept the manuscript for publication in the journal in its current form, but we would like to consider a revised version that addresses the reviewers' and editors' comments. Obviously we cannot make any decision about publication until we have seen the revised manuscript and your response, and we plan to seek re-review by one or more of the reviewers. In particular, please address in your response and through changes to the text and references, where appropriate, the concern mentioned by several of the reviewers regarding overlap between this study and recently published works (please see the first comments from Reviewers 1, 3, 4, and 5). 

We expect to receive your revised manuscript by Apr 14 2020 11:59PM. Please email us (plosmedicine@plos.org) if you have any questions or concerns.

We look forward to receiving your revised manuscript. 

Sincerely,

Caitlin Moyer, Ph.D.

Associate Editor 

PLOS Medicine

plosmedicine.org

1. Data Availability: Thank you for pointing out that your data are freely available. In your data availability statement, please state the location of data both within the paper, and also provide the links (as you have done in the supporting information section).

2. Did your study have a prospective protocol or analysis plan? Please state this (either way) early in the Methods section.

3. Title: Please revise your title to: “The relationship between circulating lipids and breast cancer risk: a Mendelian randomization study” or similar.

4. Abstract: Line 10: The word “randomization” is missing at the end of the last sentence of the background.

5. Abstract: Please define abbreviations for BCAC, SD, OR, MR at their first use. Please also give some brief context for “ABO locus”, to indicate relevance for blood group, etc.

6. Abstract: In the last sentence of the Abstract Methods and Findings section, please describe the main limitation(s) of the study's methodology.

7. Abstract: Line 27: Please change “find” to “found”.

8. Abstract: Conclusions: Please modify the first sentence of the conclusion to: “Genetically elevated plasma HDL levels appear to be associated with increased breast cancer risk” or similar. The phrase "In this study, we observed ..." may be useful.

10. Introduction: Line 56: Please define the abbreviation for GWAS at first use.

11. Introduction: Line 58: Please define the abbreviation for “ER”

12. Introduction: Line 82-83: Please remove the description of the findings from the introduction.

13. Methods: Please include a statement to generally describe that institutional ethical approval and consent were obtained for those participating in the cohorts that provided the summary/ GWAS data.

14. Methods: Line 101: Should “plink” be capitalized?

15. Methods: Line 117-119: Please make sure that the referencing of these studies/criteria follow Vancouver style: https://journals.plos.org/plosmedicine/s/submission-guidelines#loc-references

16. Methods: Line 139: please define abbreviation for LD.

17. Results: Line 151: Thank you for spelling out these abbreviations- however, please spell them out where they are first used earlier in the manuscript.

18. Results: Line 248-251: For clarity, we suggest that you first mention only the significant interactions (and the method that produced them) and then list the methods/interactions that fall short of significant second. Please explicitly clarify where significance indicates statistical significance, and please clarify what is meant by “nominal association” - does this indicate statistical significance? If so, please state that.

19. Results: Throughout the results section, where positive or negative relationships are noted, please clarify whether relationships are statistically significant, for example at Line 230: “We observed a positive relationship…” it seems such relationships reach statistical significance, while relationships described at line 248 do not. We suggest revising the sentence: “However, most P-values were not significant; the only nominal association (P < 0.05) was with total cholesterol and ER-negative breast cancer (P=0.04).” to: “However, most genetic correlation estimates between breast cancer and lipid traits were not statistically significant; the only nominal statistically significant association (P < 0.05) was with total cholesterol and ER-negative breast cancer (P=0.04).” or similar.

20. Figure 1: Some symbols in Figure 1 do not appear to be displaying properly, please check the formatting. Please define all abbreviations in the figure legend (i.e. for MR, OR, VW, HDL, LDL, TG).

21. Figure 2: Please define abbreviations “N SNPs” and “OR” in the figure legend. In the right-side panel, please describe the dotted line and bars.

22. Discussion: Line 308: Please define abbreviation for “IVS”.

23. Discussion: Line 323: Please edit the sentence to “These findings support a causal relationship between increased HDL cholesterol and increased breast cancer risk…” or similar.

24. References: For in-text citations, please use square brackets, like this [1].

25. Supporting Information: Please provide titles for supplementary figures and tables, and please ensure that all abbreviations present in the figures and tables are defined in the figure legends.

26. Supporting Information: For consistency, please use the "Vancouver" style for reference formatting, and see our website for other reference guidelines: https://journals.plos.org/plosmedicine/s/submission-guidelines#loc-references

Comments from the reviewers:

Reviewer #1: PMEDICINE-D-19-03886. 

The authors conduct a nice, thorough Mendelian Randomization analysis of the relationship between blood lipids and breast cancer risk. Similar work has been done previously (Nowak and Ärnlöv, Nat Comm 2018; Qi and Chatterjee Nat Comm 2019). Nonetheless, the work presented here constitutes a more comprehensive MR analysis than those papers, but the authors should cite the work by Qi and Chatterjee. I have a few questions for clarification

1. It is not clear to me why the authors sometimes uses lipids summary statistics from MVP, sometimes from GLGC and sometimes from the meta-analysis between the two.

2. There is some evidence of sex difference in SNP-lipids associations (e.g. Asselbergs, AJHG 2012). If possible, it would be interesting to see MR analysis using SNPs associated with lipids specifically in women, although this might not be possible if the authors do not have access to the raw GWAS and lipids data for such analyses.

3. It is not clear to me why the authors present data from iCOGS, OncoArray and GWAS separately? Unless there are explicit reasons for this, I suggest only using the meta-analysis results for breast cancer only. 

4. The authors pruned pleiotropic variants and reran the analyses similar to the Nowak paper (top page 10). Even though they note that the confidence intervals increased, it is also worth noting that the effect estimates changed quite a bit as well, so the null association is not only due to the increased confidence intervals. In particular, for TG the confidence intervals barely overlap.

5. As the authors pointed out, one of the things that distinguish this work from Nowak and Ärnlöv is the multivariable approach presented here. Another strength with this paper is the assessment of ER+ and ER- breast cancer. I think that the paper would gain from a more comprehensive analysis of ER+ and ER- breast cancer specifically, using some of the multivariable approaches described here.

6. Are the results from rho-HESS (Suppl Table 13) genetic covariances or genetic correlations? They seem small for being correlations. For the uninformed, a quick explanation about the differences in the overall (not local) genetic correlations as estimated from rho-HESS and LDSC would be helpful and help interpret the differences in results. Related to that, it would be interesting to expand on the discussion on the local genetic correlation results. Is the main conclusion from those analyses that ABO is casually associated with both blood lipids and breast cancer risk? 

7. Minor point: Refs no. 5 and 7 seem to be the same. 

Reviewer #2: This is a comprehensive Mendelian randomization study of lipids and breast cancer risk. Though I am not familiar with the more novel methods used in the study, it is clear that the authors apply both conventional and state-of-the-art methods on well-known large GWAS data to investigate the topic in depth. The study is well-written, and the authors generally use balanced language in relation to their methods and findings. I have few specific comments: 

Introduction

1. Reverse causation has been frequently debated to be involved in the association between cholesterol and risk of several forms of cancer. I suggest to mention the potential for reverse causation also in breast cancer if this is of particular relevance also for breast cancer.

2. Reference 6 is about survival amongst cancer cases and not about cancer risk, as referred to in the introduction and which the present study is about. The reference also lacks authors in the reference list. 

3. Reference 7 is not about statins, but is a meta-analysis of lipids and breast cancer risk. Reference 5 and 7 are the same.

4. In reference 9, an equally suggestive positive association between HDL and breast cancer was shown as the suggestive negative association between TG and breast cancer. Also, for TG, the authors should report the association in ref 9 to be inverse/negative. Reference 9 and 38 are the same. Reference 38 referred to as the study by Nowak et al (first sentence page 15) is wrong.

5. There are several strong risk factors of breast cancer in addition to BMI (BMI not even being amongst the most important ones) and age at menarche, so including particularly these in an MR analysis is not the "ideal approach", the authors should remove or reformulate the sentence on page 4, lines 69-71.

6. Please remove the plentiful results reported from the last paragraph of the introduction, as they are not essential to understand the aims.

Methods

1. Page 6, lines 107-108. Please revise the sentence if the lipid, BMI and age at menarche genetic associations were estimated in different cohorts to those of breast cancer genetic associations, i.e. the important part being no overlap with breast cancer cohorts.

2. Include a reference for the multivariable MR analysis performed.

Reviewer #3: This study uses genetic instruments defined in the Million Veterans Project, to assess the potential role of HDL and LDL in breast cancer risk data from the BCAC consortium.

This study ignores literature published during 2019, when 2 other papers on Mendelian randomisation provide essentially the same results. One of them, by Beeghly-Fadiel et al. on behalf of the BCAC group, had access to individual level data to stratify the analyses not only by ER status, but also for menopausal status, age, and BMI. In that study, only the genetic instrument for HDL was associated with BC. The second study, by Qi and Chatterjee also found the increased risk restricted to HDL, not to LDL or TG. The authors should revise their manuscript in light of these of the studies and explain the reason of their different findings.

The adjustment for BMI and age at menarche, using genetic instruments is an interesting approach when no observed data is accesible, but it should be commented what are the R2 of these instruments? if low, residual confounding may be high. The study bye the BCAC group analysed these associations among controls and only found consistent associations with age. Similarly, the R2 of the pruned genetic instruments finally used would be important to know. Could genetic ancestry bias the results? The qqplots of the genetic instruments in the MVP study showed some inflation.

It is not explicit if the heterogeneity pruning was performed at the level of genotyping array, or at the country/population level, since each of the 3 BCAC subsets comprises multiple smaller studies performed in diverse populations.

The discussion is very poor, with some emphasis on the potential adverse effects on increasing HDL by statins, but no discussion about the results of meta-analysis of cohort studies on BC risk that have measured plasma lipid levels.

One of the interesting analysis performed is the gene-level approach. The finding of ABO and BRCA is intriguing. The epidemiological evidence of such association has been explored previously, with a possible higher risk of the A group, but the discussion is missing.

Minor:

Abstract: end last sentence of Background

Reviewer #4: GENERAL COMMENTS

------------------

Johnson et al. conduct a thorough and comprehensive MR analysis of lipid traits and breast cancer risk. The methods applied are appropriate and well described, and their conclusions reasonable given the results. My only major concern relates to the novelty of the study, given that MR analyses of lipid traits and breast cancer risk have previously been published, including a recent analysis by Beeghly-Fadiel et al. (IJE, 2019) (although it is important to note that the paper by Beeghly-Fadiel may have been published after submission of this paper by Johnson et al., I am not sure). 

MAJOR COMMENTS

------------------

1) Beeghly-Fadiel et al (https://www.ncbi.nlm.nih.gov/pubmed/31872213) have recently published an MR analysis of lipid traits and breast cancer risk, using similar data sets and arriving at similar conclusions. The paper by Beeghly-Fadiel et al should be referenced, and given the similarity of the papers, a comprehensive comparison of methods employed and conclusions drawn should be made. 

MINOR COMMENTS

------------------

1) Line 10, "Randomization" is missing from the end of the sentence. 

2) Line 109, authors should state whether they are using an additive or multiplicative random effects model. 

3) The authors should state how they harmonized the lipid and breast cancer GWAS SNPs (I assume it was using TwoSampleMR). They should clarify that they checked that all SNP-exposure association estimates and the SNP-outcome association estimates used the same effect alleles, and that there existed no palindromic or contradictory alleles.  

4) Lines 173-174, the authors state that they also tested the relationship between lipids and breast cancer also using GLGC along. I assume this was as a sanity check of heterogeneity between the data sets, but this should be made clearer. 

5) The authors state that the meta-analysis of the Klarin and Miller GWAS "appears well-calibrated". However, the lambda values in Supplementary Figure 1 range from 1.29 to 1.51, indicating some inflation of the test statistics. The authors should state that such inflation exists, and discuss whether the inflation could affect the results of their MR analyses. 

6) How was the set of genes with protein products with key roles in HDL or LDL biology chosen? Is it from prior knowledge of the authors?

7) How evidence of directional pleiotropy was assessed should be described in the methods section. 

8) That causal effects were also estimated using MR-Egger, median-based, and mode-based approaches should be mentioned in the methods. 

9) Estimates of study power should be provided for each of the lipids. I assume that the powers for the null results are good, but it would be useful to have estimates presented anyway. 

10) Line 163, triglycerides should be abbreviated for consistency. 

11) There have been a number of RCT of statins. Have any of these RCTs observed the predicted effects on breast cancer incidence?

Alex Cornish (ICR, London). I am happy to have my name made available to the authors.

Reviewer #5: My main concern with the manuscript is that the headline findings have all already been published - based on the same or similar datasets and using Mendelian randomisation methods - and therefore this manuscript does not add anything new to the literature at all.

In this manuscript, Johnson et al reports an association between HDL cholesterol and risk for all breast cancers. However, the same association (overlapping confidence intervals and similar point estimates) has already been reported by Beeghly-Fadiel et al International Journal of Epidemiology 23 December 2019 (doi.org/10.1093/ije/dyz242) using the (nearly) identical Breast Cancer Association Consortium (BCAC) data set. Both Johnson et al and Beeghly-Fadiel et al report results taking into account the effect of body mass index, menopausal status, and estrogen receptor status. Both papers perform multivariable Mendelian randomisation analyses for all three lipid traits evaluated - LDL, HDL, TG.

In this manuscript, Johnson et al also report genome-wide genetic correlation between cholesterol levels and breast cancer. However, these have already been reported by Jiang et al. Nature Communications 25 January 2019 using the same BCAC data set used in this manuscript (https://doi.org/10.1038/s41467-018-08054-4).

Both Johnson et al and Beeghly-Fadiel et al report an inconsistent relationship between triglycerides and breast cancer risk.

The only novel results presented by Johnson et al are gene-level analyses for HDL-associated genes (Figure 2) and local genetic correlation analyses that identify pleiotropic breast cancer risk- and LDL-associated SNPs near the ABO gene. However, this region is already known (published) to harbour genome-wide significant risk SNPs for LDL cholesterol and breast cancer risk in the publicly available data sets used by Johnson et al. The ABO locus itself is established as a highly pleiotropic region and this result is therefore not surprising (see for example: 10.1038/ng.3570 ). Further, the results in Figure 2 of Johnson et al can be worked out using supplementary data from Beeghly-Fadiel et al. Finally, Johnson et al use cholesterol SNPs from the Million Veteran Program but the use of these MVP SNPs does not make any qualitative difference to the results compared to the use of the SNPs from the Global Lipid Genetics Consortium (used by Beeghly-Fadiel et al).

[LINK]

---

## [Decision Letter · Decision Letter 1]

11 May 2020

Dear Dr. Voight,

Thank you very much for submitting your revised manuscript "The relationship between circulating lipids and breast cancer risk: a Mendelian randomization study" (PMEDICINE-D-19-03886R1) for consideration at PLOS Medicine. 

Your paper was evaluated by a senior editor and discussed among all the editors here. It was also discussed with an academic editor with relevant expertise, and sent back to two of the reviewers for re-review. The comments from the academic editor are pasted below. Both of the reviewers were satisfied with the revised version and noted no further comments.

In light of the comments from the academic editor, I am afraid that we will not be able to accept the manuscript for publication in the journal in its current form, but we would like to consider a revised version that addresses the comments. Obviously we cannot make any decision about publication until we and the academic editor have seen the revised manuscript and your response. 

In revising the manuscript for further consideration, your revisions should address the specific points made by the editors and the academic editor. Please also check the guidelines for revised papers at http://journals.plos.org/plosmedicine/s/revising-your-manuscript for any that apply to your paper. In your rebuttal letter you should indicate your response to the reviewers' and editors' comments, the changes you have made in the manuscript, and include either an excerpt of the revised text or the location (eg: page and line number) where each change can be found. Please submit a clean version of the paper as the main article file; a version with changes marked should be uploaded as a marked up manuscript.

We expect to receive your revised manuscript by May 25 2020 11:59PM. Please email us (plosmedicine@plos.org) if you have any questions or concerns.

We look forward to receiving your revised manuscript. 

Sincerely,

Caitlin Moyer, Ph.D.

Associate Editor 

PLOS Medicine

plosmedicine.org

1. Abstract: Line 18: Please use either Mendelian randomization or the abbreviation MR consistently.

2. Abstract: Line 29: Please revise this sentence to avoid implications of causality: “In addition, we found evidence that genetic variation at the ABO locus is associated with both lipid levels and breast cancer” or similar.

3. Abstract: Conclusions: Line 35: Would it make sense here to also mention the association between LDL and breast cancer risk that you identified with multivariable MR? (mentioned above at Line 22-23)?

4. Author Summary: Thank you for providing an author summary. Under “Why was this study done?” we suggest that you combine the first two bullet points. At line 35, under the third bullet point, we suggest changing “genetics” to “Mendelian randomization methods” or similar to enhance specificity.

5. Author Summary: Please use the first person perspective. For example, please use “We tested…” at line 49.

6. Under “What do these findings mean?” Please delete the first and third bullet point, and replace it with a sentence that can address the study implications without overreaching what can be concluded from the data; we suggest: "Further research will be needed to investigate the possibility that manipulation of LDL or HDL levels can influence risk of breast cancer"

7. Introduction: Line 93: Rather than "nominal", please change to "association of uncertain significance" or similar, to clarify.

8. Introduction: Line 101-105: This would be more appropriate in the Discussion section: “A recent study by Beeghly-Fadiel et al., published following the submission of this manuscript, performed an MR analysis of breast cancer risk that considered potentially confounding risk factors and draws some of the same major conclusions as we do [21]. We highlight the distinctions between that study and the present manuscript in the Discussion.”

9. Methods: Lines 133-135: Please provide details on the ethical approval/name the specific University IRBs that provided approval. Please specify whether informed consent was written.

10. Discussion: Lines 345-346: We suggest revising this sentence to read: “Using Mendelian randomization, we provide evidence that genetically elevated HDL and LDL levels are associated with increased risk for breast cancer…”

11. Discussion: Lines 371-372: We suggest not abbreviating Beeghly-Fadiel et al. “(hereafter B-F)” as this is only used once in the paragraph, and could cause confusion.

12. Checklist: Please ensure that the study is reported according to the STROBE guideline, and include the completed STROBE checklist as Supporting Information (or, please report your study according to the relevant guideline, which can be found here: http://www.equator-network.org/) 

Please add the following statement, or similar, to the Methods: "This study is reported as per the Strengthening the Reporting of Observational Studies in Epidemiology (STROBE) guideline (S1 Checklist)."

13. Supplementary Tables: Please provide titles and legends for each individual table and figure in the Supporting Information, rather than including these as a list in another supporting information file.

Comments from the Academic Editor:

This paper uses Mendelian randomization to assess the causal effects of lipids on breast cancer, but a much stronger case (in Introduction and Discussion) needs to be made to clarify what it adds to the existing knowledge from similar MR studies. The analyses reported are very comprehensive, and yet the rationale and practical implications of the different approaches used is not always clear to the reader – which makes it difficult to judge its added value compared with previous MR work. For example, the authors don’t discuss the implications of using a gene-specific vs. a conventional MR analysis, or a local vs. genome-wide genetic correlation analysis, and why using a gene-specific MR or a local genetic correlation analysis is important. Moreover, the MR and genetic correlation approaches answer different questions, and yet this is not clearly discussed. 

Regarding the MR methods used, I think some justification (or reference to a method paper) is required for the method used to perform the “Heterogeneity analyses for single trait MR” described in the Supplement. Similarly, more justification is needed for the pruning used in the multivariable analysis (“stepwise post-hoc procedure to remove genetic instruments that contributed the most to QA”). The tests of instrument strength and validity reported in the Supplement also require some mention to correlation assumptions when applied to two-sample MR.

Comments from the reviewers:

Reviewer #4: The authors have addressed my comments.

Alex Cornish (ICR, London).

[LINK]

---

## [Editor Report · Decision Letter 2]

6 Jul 2020

Dear Dr. Voight,

Thank you very much for re-submitting your manuscript "The relationship between circulating lipids and breast cancer risk: a Mendelian randomization study" (PMEDICINE-D-19-03886R2) for review by PLOS Medicine.

I have discussed the paper with my colleagues and the academic editor. I am pleased to say that provided the remaining editorial and production issues are dealt with we are planning to accept the paper for publication in the journal.

The remaining issues that need to be addressed are listed at the end of this email. Please take these into account before resubmitting your manuscript. In particular, please be sure to fully address the point below (point #1) regarding MR and homogeneity assumptions.

[LINK]

In revising the manuscript for further consideration here, please ensure you address the specific points made by the editors. In your rebuttal letter you should indicate your response to the editors' comments and the changes you have made in the manuscript. Please submit a clean version of the paper as the main article file. A version with changes marked must also be uploaded as a marked up manuscript file.

We look forward to receiving the revised manuscript by Jul 13 2020 11:59PM. 

Sincerely,

Caitlin Moyer, Ph.D.

Associate Editor 

PLOS Medicine

plosmedicine.org

Requests from Editors and the Academic Editor:

1. Please address the following note from the academic editor regarding Lines 177-179 in the Methods: 

[The authors have addressed all the comments, but there is one sentence in Methods about multivariable MR which is misleading - p. 9, first paragraph: “Because MR assumes homogeneity between each instrument’s exposure to outcome effect, we then tested for instrument strength and validity [35], and removed instruments driving heterogeneity (Supplementary Methods).”. 

In reference [35] that they cite, Sanderson et al. talk about: a) “good” heterogeneity – where a modified version of Cochran’s Q statistic is used to assess instrument strength; and b) “bad” heterogeneity – where the Cochran’s Q statistic is used for testing instrument validity (e.g. due to pleiotropy), similarly to what proposed for classical (single-exposure) MR (Greco M, Del F, Minelli C, Sheehan NA, Thompson JR. Detecting pleiotropy in Mendelian randomisation studies with summary data and a continuous outcome. Stat Med 2015;34: 2926–40). So it’s unclear what “Because MR assumes homogeneity between each instrument’s exposure to outcome effect” means in the sentence above. I guess what it means is that MR assumes homogeneity in the ratio of the genetic effect on the outcome on the genetic effect on the exposure (Wald estimator) across instruments, which means that all instruments are assumed to be valid - but if this is the case, the whole sentence needs to be re-written.]

2. Data analysis plan: Please update the link for the BCAC summary data: http://bcac.ccge.medschl.cam.ac.uk/bcacdata/oncoarray/gwas-icogs-and-oncoarray-summary-results/

3. Abstract: Line 14 (and anywhere else): Please replace "subject" with participant, patient, individual, or person.

4. Introduction: Lines 108-111: The term “nominal” is used twice here, and the meaning is not clear. If you mean “not statistically significant” please make that clear. Please revise the sentence using a more specific term to indicate your meaning: “This study did not find a nominal association between any lipid trait and breast cancer using lipid summary statistics, though a previous study with a smaller breast cancer GWAS sample size did report a nominal negative genetic correlation between triglycerides and breast cancer risk [16].

5. Methods: Lines 142-144: Please provide the references for these two datasets again here. “For the Willer et al and BCAC datasets, we refer the reader to the primary GWAS manuscripts and their supplementary material for details on consent protocols for each of their respective cohorts.”

6. Discussion: Lines 398-400: Similar to the above comment, can you please clarify the meaning of “nominal” in this sentence: Third, while both studies are consistent in their relationship between HDL and BC, we reported a nominal association with LDL levels when considering all risk and confounding factors jointly.”

7. Checklist: Thank you for including the STROBE-MR extension. We agree this would be the appropriate checklist for your study; however, can you please also include the STROBE checklist, as the STROBE-MR is still preliminary/unpublished?

8. References: Please use the "Vancouver" style for reference formatting, and see our website for other reference guidelines https://journals.plos.org/plosmedicine/s/submission-guidelines#loc-references

A few citations seem to be missing information: (e.g. # 9, 10, 29, 47)

9. Supporting information figures 4-9 and 13: Rather than indicating significance with *p<0.05 and **p<0.001, please report the exact p values and 95% CIs associated with the ORs.

[LINK]

---

## [Editor Report · Decision Letter 3]

10 Aug 2020

Dear Dr. Voight, 

On behalf of my colleagues and the academic editor, Dr. Cosetta Minelli, I am delighted to inform you that your manuscript entitled "The relationship between circulating lipids and breast cancer risk: a Mendelian randomization study" (PMEDICINE-D-19-03886R3) has been accepted for publication in PLOS Medicine. 

PRODUCTION PROCESS

PRESS

PROFILE INFORMATION

Thank you again for submitting the manuscript to PLOS Medicine. We look forward to publishing it. 

Best wishes, 

Caitlin Moyer, Ph.D.

Associate Editor 

PLOS Medicine

plosmedicine.org